# Targeted Quantification of Protein Phosphorylation and Its Contributions towards Mathematical Modeling of Signaling Pathways

**DOI:** 10.3390/molecules28031143

**Published:** 2023-01-23

**Authors:** Panshak P. Dakup, Song Feng, Tujin Shi, Jon M. Jacobs, H. Steven Wiley, Wei-Jun Qian

**Affiliations:** Biological Sciences Division and Environmental Molecular Sciences Laboratory, Pacific Northwest National Laboratory, Richland, WA 99352, USA

**Keywords:** targeted phosphoproteomics, LC-SRM, quantification, phosphorylation, signaling, mathematical modeling, cancer

## Abstract

Post-translational modifications (PTMs) are key regulatory mechanisms that can control protein function. Of these, phosphorylation is the most common and widely studied. Because of its importance in regulating cell signaling, precise and accurate measurements of protein phosphorylation across wide dynamic ranges are crucial to understanding how signaling pathways function. Although immunological assays are commonly used to detect phosphoproteins, their lack of sensitivity, specificity, and selectivity often make them unreliable for quantitative measurements of complex biological samples. Recent advances in Mass Spectrometry (MS)-based targeted proteomics have made it a more useful approach than immunoassays for studying the dynamics of protein phosphorylation. Selected reaction monitoring (SRM)—also known as multiple reaction monitoring (MRM)—and parallel reaction monitoring (PRM) can quantify relative and absolute abundances of protein phosphorylation in multiplexed fashions targeting specific pathways. In addition, the refinement of these tools by enrichment and fractionation strategies has improved measurement of phosphorylation of low-abundance proteins. The quantitative data generated are particularly useful for building and parameterizing mathematical models of complex phospho-signaling pathways. Potentially, these models can provide a framework for linking analytical measurements of clinical samples to better diagnosis and treatment of disease.

## 1. Introduction

Protein phosphorylation is the most common and widely studied post-translational modification (PTM) in eukaryotes. Phosphorylation is involved in numerous signal transduction processes including proliferation, differentiation, apoptosis, and metabolic processes that maintain cellular homeostasis and direct physiological responses to stimuli [1]. Phosphorylation regulates function by multiple mechanisms, including conformational changes that alter protein structure and creating docking sites for the assembly of multi-protein complexes [2]. Kinases introduce negatively charged phosphate groups which interact with neutral hydroxyl groups on Serine (Ser), Threonine (Thr), and Tyrosine (Tyr) residues, resulting in a polar anionic moiety. Conversely, phosphatases reverse this biochemical process with the removal of the phosphate group. The dynamics of phosphorylation by these two counteracting enzymes are tightly controlled to properly propagate signal transduction events and modulate protein activity in each biochemical cascade [1,2,3,4,5]. An aberration of protein phosphorylation within any network can cause adverse effects that manifest in many disease states including cancer [6,7,8], diabetes [9,10,11], Alzheimer’s disease [12,13], rheumatoid arthritis [14,15], muscular dystrophy [16] and cardiovascular diseases [17,18]. Furthermore, multiple treatment strategies target protein kinases such as: Imatinib, a tyrosine kinase inhibitor for chronic myelogenous leukemia [19]; Duvelisib, a PI3K inhibitor for chronic lymphocytic leukemia and follicular lymphoma [20,21]; Larotrectinib, a NTRK inhibitor for solid tumors [22]; and Vemurafenib, a BRAF serine/threonine kinase inhibitor for melanoma [23], demonstrating the importance of signal transduction in cancer [24]. In addition, measuring protein phosphorylation is becoming centrally important to understanding healthy states, disease pathogenesis, and identify treatment strategies. Hence, identifying and quantifying protein phosphorylation within the context of complex phosphorylation-mediated signaling cascades has garnered substantial interest over the last decade.

Although substantial progress has been made in elucidating signaling events and uncovering therapeutic strategies, many challenges remain in understanding the relationship between signaling dynamics and disease progression. Specifically, the functions of signaling networks are based on the spatiotemporal dynamics that emerge from extensive protein groups and interactions collectively, rather than the simple presence or absence of individual proteins or interactions [25,26,27]. Thus, understanding signaling dynamics requires the quantitative measurements of multiple sites on multiple proteins through time and space. Advancements in the field of phosphoproteomics have allowed identification and quantification of thousands of phosphorylation sites [28,29,30,31]. Targeted phosphoproteomics has enabled the identification of site-specificity, localization of phosphorylation, the extent of phosphorylation, and quantification of absolute abundance with high throughput. This information can help to not only elucidate the biological relevance of proteins but also provide the high-resolution data needed to parameterize predictive mathematical models [32].

Computational modeling is also becoming increasingly important in understanding the collective behaviors of signaling proteins which are difficult to comprehend intuitively. Many different modeling paradigms and inference algorithms have been proposed to facilitate modeling, simulation, and inference of different types of biological systems [33]. In particular, the kinetic modeling of reaction networks and its derivative modeling method, rule-based modeling, are designed for signaling networks [34,35]. Modeling and experimental approaches are complementary and need to be integrated so that phosphorylation data can inform model structures and parameters while models, in turn, can guide experimental designs. Therefore, it is necessary to consider the information required for different types of models when designing phosphoproteomics experiments. In this review, we discuss the challenges facing targeted phosphoproteomics, the analytical advancements made to address those challenges, and the prospects of using targeted phosphoproteomics data to build predictive models of signaling pathways for biomedical applications.

## 2. The Case for Measuring Protein Phosphorylation by Mass Spectrometry

Current research in protein phosphorylation is focused on identifying phosphorylation sites, quantifying their abundance, and characterizing functional impact. From an analytical standpoint, there are inherent challenges in phosphorylation analyses due to the labile nature of the modification and sub-stoichiometric concentrations [36]. While different biochemical approaches have been developed and utilized to study protein phosphorylation, they all have inherent limitations. A few common examples are phosphoimaging and immunoassays. Historically, phosphoimaging initially depended on the use of radioisotopes, such as the incorporation of the ^32^P-labeled ATP to monitor kinase activity towards specific substrates [37]. More recently, fluorescent-based dyes have been used to stain for phospho-residues, selectively or universally, by direct imaging on blotting membranes [38,39]. Alternatively, antibody-based immunoassays like immunoblotting and enzyme-linked immunosorbent assay (ELISA) have been used as a substitute to radiolabeling techniques and can allow for signal amplification [40]. Unfortunately, the above-mentioned methods are generally not site-specific and do not provide stoichiometry information of individual phosphosites [40]. Consequently, the emerging mass spectrometry (MS)-based technologies aim to address these limitations of sensitivity, specificity, and rigor in measuring protein phosphorylation in complex biological samples.

### MS Analysis of Protein Phosphorylation

The development of Mass Spectrometry (MS)-based proteomics has revolutionized the study of protein phosphorylation. Conceptual and technical breakthroughs in MS-based phosphoproteomics have enabled the detection and quantification of tens of thousands of phosphosites per sample [30,41,42]. Generally, proteins are digested to produce peptides that are separated by reversed-phase liquid chromatography and analyzed by mass spectrometry (LC-MS). Shotgun proteomics allows the discovery of phosphorylation sites through a data-dependent acquisition (DDA) mode. The DDA mode relies on a full-scan MS spectrum determination of peptide species present, then MS/MS acquisition of a list of most intense peptide precursor ions [43]. DDA approaches are suitable for exploring complex biological samples and identifying large numbers of phosphopeptides without prior knowledge [44]. Therefore, DDA is responsible for the identification of hundreds of thousands of phosphorylation sites across multiple specimens and species [45]. Such site-specific phosphorylation information has been collated into publicly available resource databases for use by the research community. PhosphoSitePlus [46] is the most comprehensive database with over 290,000 non-redundant phosphosites, while PHOSIDA [47,48], Phospho.ELM [49,50], and PhosphoPEP [51,52] contain over 70,000, 40,000, and 10,000 phosphosites, respectively, and counting. However, DDA is inherently biased towards more abundant phosphopeptides, which means that a phosphopeptide of interest is often not detected. In contrast, targeted methods of phosphoproteomics provide a more reliable way to monitor a specified number of ‘target’ phosphoproteins within and across biological samples. This facilitates hypothesis-driven research while simultaneously enabling high specificity and accurate quantification.

## 3. General Workflow for Targeted Phosphoproteomics

The general workflow for targeted phosphoproteomics is outlined in Figure 1. Typically, it involves the proteolytic digestion of proteins and post-digestion manipulations, such as enrichment to enhance recovery of phosphopeptides followed by MS analysis using a reference internal standard.

### 3.1. Surrogate ‘Target’ Phosphopeptide Selection and Enzymatic Digestion

The careful selection of target peptides is a crucial step in the identification and quantification of target proteins. The primary criteria for selection of surrogate peptides are that they must uniquely represent the protein of interest and be experimentally detectable within a proteome. They are often referred to as ‘proteotypic peptides’ [55,56]. Proteotypic peptides should be quantifiable (or quantotypic) and correlate with levels of the parent protein [55,57]. However, the selection of quantitative surrogate phosphopeptides for targeted analysis can be challenging because it must include the phosphorylated sites of interest. The typical approach of protein digestion using proteases can complicate the situation. Ideally, the enzyme(s) of choice should have efficient access to the cleavage sites. However, when phosphopeptides contain phosphorylated sites adjacent to the cleavage residues, they can form salt bridges which inhibit the activity of the enzyme and results in missed cleavages [58]. Trypsin is the most commonly used enzyme, which cleaves specifically at Lys and Arg C-terminal residues (except with the presence of Pro at the position after the cut site). Hence, the options for surrogate phosphopeptides can be limited. There are a few options to select alternative peptides. First, utilizing other proteolytic enzymes such as LysC (cleaves Lys C-terminal residue), AspN (cleaves N-terminal aspartic and glutamic acid residues), ArgC (cleaves C-terminal of Arg residues, including sites next to Pro), and GluC (cleaves C-terminal aspartic and glutamic acid residues) provides different options for cleavage sites. Multiple enzyme digests can be performed in parallel or sequentially to diversify peptide pools and increase coverage of the phosphoproteome. For example, sequential digestion using Glu-C and Lys-C enzymes prior to trypsin digestion has been shown to increase phosphosite identification by 83% and 40%, respectively [59,60]. Optimization of digestion efficiency, including the enzyme-to-substrate concentration ratio, optimal buffer conditions, and digestion time will be helpful with achieving reproducibility and accurate quantification.

### 3.2. Enrichment

Ongoing analytical challenges for targeted analyses include high peptide lability, sub-stoichiometric concentrations, heterogenous iso-forms, and low mass spectrometric response to protein phosphorylation [36]. It is estimated that at least 75% of the proteome can be phosphorylated, with kinases and phosphatases accounting for 1–2% of the genome [42,61]. In addition, the approximate relative abundance for pSer/pThr/pTyr species is 1800:200:1 [28]. Thus, phospho-tyrosine residues, which are critical signatures for tyrosine receptor kinase activation, constitute a very small fraction of all generated peptides. Hence, the efficient enrichment of pSer, pThr, and pTyr residues is crucial for robust and reliable phosphoproteome analyses. Towards this end, chemical resin- and antibody-based enrichment strategies have been developed and applied to enrich either total or tyrosine-phosphorylated phosphopeptides prior to MS analysis.

#### 3.2.1. Immobilized Metal Affinity Chromatography (IMAC)

The most common phosphoenrichment technique introduced over three decades ago is IMAC. It is based on the principle that positively charged metal ions such as Fe^3+^, Ga^3+^, Ti^4+^, and Zr^4+^ can chelate and capture negatively charged phosphopeptides [62,63,64,65]. These metal ions are immobilized on resins that are commonly coated with iminodiacetic acid (IDA) or nitrilotriacetic acid (NTA) [66]. However, IMAC workflows can show reduced sensitivity caused by buffer interference [67] and its specificity can be reduced by acidic peptides that contain multiple aspartic acids and glutamic acids [68]. Though methyl esterification of the acidic amino acid residues has been used to prevent non-specific binding, it introduced more biological complexity as those modified residues can be prone to spontaneous deamidation and subsequent methylation [69,70]. An alternative approach is the complete protonation of acidic amino acid residues with acid at a low pH (below 2) [62]. It is also important to remove all traces of nucleic acids during sample protein extraction. These can competitively bind to the IMAC matrix, thus reducing IMAC efficiency [71].

To increase specificity and improve robustness of phosphoenrichment, several modifications have been applied to IMAC methodologies. Elements like Nickel have been linked to NTA-based resins to improve specificity [72]. Metal (IV) phosphate/phosphonate chemistry has revealed stronger coordination of metals like Zr^4+^ and Ti^4+^ with phosphates and carboxylic acid groups, and this has been implemented in approaches like the Ti^4+^-IMAC technology [63,73]. The preferential affinity of different metals towards specific phosphopeptides has led to sequential enrichment strategies. For example, Ti^4+^-IMAC has been shown to be able to enrich multiple basic amino acid residues in addition to acidic residues [74]. Another example is the sequential use of Ga^3+^-Fe^3+^-IMAC where Ga^3+^ is initially applied to enrich multi-phosphorylated peptides followed by Fe^3+^ to capture monophosphorylated peptides [75]. The extensive optimalization of IMAC-based enrichment strategies has significantly improved their usability for targeted quantification.

#### 3.2.2. Metal Oxide Affinity Chromatography (MOAC)

MOAC enrichment is based on the Lewis acid-base interaction principle where metal oxides and hydroxides interact with oxygen atoms of phosphate groups to form complexes. Compounds like TiO_2_, ZrO_2_, Fe_3_O_4_, Ga_2_O_3_, Nb_2_O_3_, SnO_2_, Ta_2_O_5_, HfO_2_, and Al(OH)_3_ have been tested as matrixes for enrichment of phosphopeptides [76]. The first application of MOAC trapped phosphopeptides under acidic conditions on a TiO_2_-based precolumn. These were subsequently eluted in alkaline buffers for LC-MS/MS analysis [77]. The buffer conditions used in MOAC significantly influence its selectivity for phosphopeptides. For example, the addition of 2.5-dihydroxybenzoic acid (DHB) to loading and washing buffers reduces nonspecific binding of non-phosphorylated peptides to TiO_2_ and thus, increased the binding affinity for phosphopeptides [70,78]. However, residual DHB interferes with LC-MS/MS analysis. Alternatively, aliphatic hydroxy acid modified metal oxide chromatography (HAMMOC) using lactic acid and β-hydroxypropanoic acid in place of DHB improved phosphopeptide selectivity [78]. Furthermore, sequential elution on a pH gradient has been used to increase enrichment efficiency. For example, trapped mono-phosphopeptides in a TiO_2_ microcolumn can be first eluted with acidic methylphosphonate buffer followed by successive basic elution of multi-phosphopeptides with disodium hydrogen phosphate, ammonium hydroxide, and pyrrolidine [79].

Since IMAC and MOAC possess different affinities for phosphopeptides, it can be beneficial to combine both enrichment strategies. IMAC and MOAC typically favor multi-phosphorylated peptides and mono-phosphorylated peptides, respectively [67,80]. One combined approach is sequential enrichment with immobilized metal affinity chromatography (SIMAC). Here, the sample is first enriched with Fe^3+^-IMAC and multi-phosphopeptides are eluted in alkaline conditions, followed by TiO_2_ enrichment of the unbound acidic fraction for mono-phosphopeptides [81,82]. Like SIMAC, sequential enrichment using metal oxide affinity chromatography (SMOAC) offers a reverse approach. The samples are first enriched by TiO_2_ resin followed by Fe^3+^-IMAC [83]. Combining IMAC and MOAC strengthens binding selectivity and reduces the amount of starting material while deriving two distinct phosphopeptide fractions for analysis.

#### 3.2.3. Polymer-Based Metal Affinity Capture (PolyMAC)

PolyMAC is another adaptation of IMAC but using soluble nanopolymers instead of solely metal ions. The nanopolymers are dentrimers multifunctionalized with metal ions that bind to phosphopeptides in solution. The PolyMAC-phosphopeptide complexes are subsequently recovered on a solid phase by coupling to hydrazine-agarose gels, followed by washing and phosphopeptide elution. The PolyMAC-Ti formulation has been well studied and appears to display high recovery, reproducibility, and enrichment compared to IMAC and TiO_2_ methods [84,85,86].

#### 3.2.4. Antibody-Based Enrichment

Many traditional biochemical separation approaches are based on antibodies binding to specific epitopes. This has been applied to phosphoproteomics using antibodies raised against phosphorylated serine, threonine, and tyrosine residues. The metal-based methods discussed above are efficient for pSer and pThr residues, but not pTyr due to their significantly low relative abundance [87]. Fortunately, highly specific phosphotyrosine antibodies have proven to be effective in capturing the tyrosine phosphorylated peptides and proteins [88,89,90,91]. In a first application, selective immunoprecipitation of tyrosine phosphorylated proteins using anti-phosphotyrosine antibodies was able to identify known tyrosine phosphorylated proteins as well as novel sites in the epidermal growth factor (EGF) receptor signaling pathway [92]. However, most anti-phosphoserine and threonine antibodies do not bind efficiently to their respective sites and hence are not widely used in phosphoproteomics. Moreover, the need for site-specific antibodies makes immunoprecipitation unsuitable for wide coverage studies of the phosphoproteome. Nevertheless, the specificity and selectivity of antibodies provides a useful strategy when targeting specific low abundant proteins, as in the case of tyrosine-phosphorylated proteins. Furthermore, antibody- and IMAC-based strategies can be combined in a single workflow for deeper coverage of the phosphoproteome [93].

### 3.3. Fractionation

Fractionation is a step employed to reduce sample complexity either prior to phosphoenrichment or MS analysis. These strategies separate complex peptide samples into fractions that are more suitable for further enrichment or to enhance coverage depth of the phosphoproteome [94].

Strong cation exchange (SCX) and strong anion exchange (SAX) are chromatography techniques that fractionate peptides according to their charge. In SCX, positively charged cations interact with the negatively charged SCX matrix. The pH of the loading and elution buffers determine the binding and elution of different peptides. Acid buffers (pH 2.7) are used to protonate the C-terminal arginine and lysine residues and N-terminal amino acid groups, separating phosphorylated peptides from non-phosphorylated peptides by net charge states [95]. Bound peptides are eluted by increasing the ionic strength and/or pH to maximize collection of phosphopeptides in early fractions [96]. In contrast, SAX chromatography depends on the interaction of negatively charged peptides with a positively charged matrix. It is carried out using neutral-to-alkaline buffers so that the C-terminal carboxylic group and glutamate and aspartate residues are deprotonated. Most phosphopeptides are collected in later fractions by increasing ionic strength or decreasing the pH of the eluting buffer [97]. The applications of SCX and SAX are complementary as SCX favors retainment of mono-phosphorylated peptides while SAX favors multi-phosphorylated peptides [95,98]. However, selectively is an issue as other peptides with the same net charge states cannot be distinguished from phosphopeptides [94].

High pH reversed-phase liquid chromatography (HpH RPLC) is another fractionation strategy based on hydrophobic interactions [99]. It is carried out in a high pH environment and orthogonal separation is achieved by low to high pH gradient in the mobile phase [100]. HpH RPLC offers a higher resolving power compared to SCX and SAX and can be coupled with low pH LC-MS/MS [101]. In offline mode RPLC also offers the option of pooling fractions from different parts of the gradient, thus reducing the number of instrument runs for analysis [101,102]. To advance this strategy our group developed PRISM (high-pressure, high-resolution separation with intelligent selection and multiplexing) [103]. Here, high pH reversed-phase capillary LC is used to fractionate samples. Then an online SRM system monitors heavy isotope-labeled internal standards to select target fractions for downstream LC-SRM. Target fractions eluted at different times can be concatenated to increase throughput. The PRISM workflow provides a more-sensitive targeted proteomics workflow that does not require affinity enrichment [104]. RPLC fractionation prior to phosphopeptide enrichment by IMAC or MOAC has yielded extensive coverage of the phosphoproteome [105,106,107,108].

Other fractionation strategies like hydrophilic interaction chromatography (HILIC) and electrostatic repulsion-hydrophilic interaction (ERLIC) are based on hydrophilic interaction [109,110]. HILIC relies on hydrogen bond interaction between a neutral, hydrophilic stationary phase and peptides. More hydrophilic peptides are retained longer and eluted by increasing the polarity of the mobile phase, which is opposite to RPLC [109]. Alternatively, ERLIC is a subset of HILIC which utilizes both hydrophilic interactions and electrostatic forces. In the presence of high-percentage organic mobile phase at low pH, hydrophilic interactions dominate over electrostatic forces, making phosphopeptides less repulsed (or more retained) by the column compared to non-phosphorylated peptides. However, the absence of hydrophilic interactions leads to electrostatic repulsion of phosphopeptides at low pH [110,111].

## 4. Targeted MS Analysis of Phosphorylation

Current targeted methods of phosphoproteomics are based on selected reaction monitoring (SRM, also called multiple reaction monitoring (MRM)) [55] to provide precise quantification, reliable reproducibility, and robust sensitivity of ‘target’ proteins/phosphoproteins in biological samples. As technological advancements in instrumentation continue, there are targeted acquisition methods like parallel reaction monitoring (PRM) and data independent acquisition (DIA)-MS strategies that are gaining relevance. Due to the sub-stoichiometric nature of phosphorylation events, the continuous gain in sensitivity of targeted MS acquisition tools is useful to quantitatively and consistently measure target phosphopeptides. The current methods will be discussed subsequently.

### 4.1. SRM/MRM

Selected Reaction Monitoring (SRM), also known as Multiple Reaction Monitoring (MRM), is an approach for reliably measuring phosphoproteins in a highly reproducible manner within and across laboratories and instrument platforms [112,113]. Implementation of SRM/MRM to quantify phosphorylation requires that the mass shifts introduced by phosphorylation are known. MS/MS spectra are isolated and detected by linear collision-induced dissociation (CID) on traditional triple quadrupole (QQQ) mass spectrometer platforms. The first quadrupole (Q1) filters and selects predefined precursors, the second quadrupole (Q2) induces fragmentation of the species via CID, and the third quadrupole (Q3) detects the resulting product/fragment ions with high speed and sensitivity [114]. Fragment ion analysis occurs in different scan modes. In the precursor ion scan mode, only phosphate (PO_3_^−^)-modified fragments are detected in Q3 by the addition of 79 Da, which is only ideal for highly stable phosphopeptides. Alternatively, neutral loss scan mode detects the loss of a neutral group (H_3_PO_4_) from a phosphorylated peptide after low-energy CID. The loss of H_3_PO_4_ results in a mass shift of 98 Da, 49 Da and 32.7 Da for singly, doubly, and triply charged phosphopeptides, respectively. However, tyrosine-specific phosphopeptides are more stable and show no neutral loss, thus Q3 scans for an immonium ion (216.042 Da) marker [115]. The acquired mass-to-charge (*m/z*) values of precursor-fragment ion pairs monitored by Q1 and Q3 are called ‘transitions.’

In establishing SRM assays, selecting optimal transitions for each peptide is important. First, it is best practice to select the most intense fragments because the intensities of fragment ions derived from one precursor differ substantially. Low-energy CID induces the generation of b/y series fragment ions. In this fragmentation environment, y-ion signals are more common, robust, and stable compared to b-ion signals [116]. Even though y-ions are the preferred fragments for building transitions, b-ions can also be good candidates depending on the peptide. It is also good practice to exclude low *m/z* fragment ions due to low specificity and ensure that chosen fragments have *m/z* values above the precursor for high selectivity and spectral noise reduction [55]. In general, at least three transitions per phosphopeptide are usually selected for SRM analysis, to allow for room to include more targets in one assay. Once transitions are finalized, hundreds of phosphopeptides can be targeted in a single LC-MS/MS analysis. Targeted data analysis is typically performed using the Skyline (Figure 1), the most commonly used software for targeted proteomics data analysis, and other tools such as MRMer [117,118].

The first utilization of SRM for large-scale targeted phosphoproteomics was reported by White’s group in 2007 [119]. They were able to quantify phosphorylation changes in the Epidermal Growth Factor Receptor (EGFR) signaling network upon Epidermal Growth Factor (EGF) stimulation. Using a two-step enrichment procedure with selective antibody immunoprecipitation of phosphotyrosine-containing peptides followed by IMAC, they quantified 222 phosphorylated peptides across seven time points, in a single two-hour analysis.

For absolute quantification workflows, addition of chemically synthesized heavy isotope-labeled peptides containing the phosphorylated sites of interest is the gold standard for analyzing phosphopeptides and their stoichiometries [55,112,120]. These peptide analogs enable the identification of phosphopeptide targets with high confidence with reduced technical variation and false positives that could arise from co-eluting isoforms [44,53]. The retention time and ion fragmentation pattern are identical between heavy isotope-labeled and endogenous phosphopeptides, though distinguishable by a set mass difference. The intensity of the endogenous peptide relative to the added heavy isotope-labeled standard is the basis for quantification [55]. It is important to adjust the concentrations of spiked-in heavy peptides in order that the relative ratios of endogenous-to-heavy signals within a reasonable range for reliable quantification. To quantify differences between native and phosphorylated peptides, two internal standard peptides can be used: a non-phosphorylated and a phosphorylated peptide. The relative intensities of non-phosphorylated and phosphorylated peptides determine the percent change in phosphorylation at a given site which, in turn, can provide a measure of stoichiometry. Gerber et al. demonstrated this concept by quantifying the cell cycle-dependent Ser-1126 phosphorylation of Human Separase [121]. This approach can be integrated as a normalization strategy because changes in up to 25% of phosphopeptides can be attributable to varying protein levels [122]. With the complexities of biological mixtures, normalization strategies to correct for systematic biases are also often critical [123].

In general, standard-based quantification for targeted phosphoproteomics is gaining relevance with increasing experimental demonstrations by different groups (Table 1). The largest scale utilization of this approach so far has been by Carr’s group, quantifying 284 phosphosites in a multiplexed quantitative assay called SigPath [93]. Our group has developed assays to measure absolute abundances of proteins and phosphorylation sites over time in the Epidermal Growth Factor Receptor-Mitogen-Activated Protein Kinase (EGFR-MAPK) pathway to better understand rewiring in drug resistant cancers [32,53]. We have found that current AQUA tools can reliably measure changes in the phosphoproteome, and hold promise for more widespread research and modeling applications.

### 4.2. PRM

In the ongoing effort to avoid limiting the number of targets, technical advancements in high-resolution mass-measuring instruments like the hybrid quadrupole-orbitrap mass spectrometer has enabled the development of Parallel Reaction Monitoring (PRM) [130]. Like SRM, precursor ions of interest are selected and fragmented in Q1 and Q2, respectively. However, unlike SRM the Orbitrap replaces the Q3 to scan all fragment ions at high resolution, enabling the selection of the best transitions post-acquisition [131,132]. The Quadrupole-Orbitrap instrumentation setup also offers trapping capabilities which are valuable in analyzing low abundance events like phosphorylation. This approach reduces potential background interference usually encountered with complex biological samples, offering both qualitative capabilities and quantitative analysis [130].

PRM data acquisition takes advantage of certain Quadrupole-Orbitrap instrumentation parameters that can be optimized to improve analysis of complex samples. The quadrupole isolation width can be narrowed to directly control the selectivity of precursor ions and exclude interferences, and the trapping fill time can be longer to enhance sensitivity and widen dynamic detection range [130]. PRM has been used in a highly reproducible manner to measure 96 reduced-representation probes in ~200 samples across cell types, treatment conditions, and timepoints [126]. There are also a few small-scale studies that have used PRM to confirm and quantify protein targets from previous experimental observations [133,134,135]. A more recent data acquisition method, called internal standard triggered-parallel reaction monitoring (IS-PRM), uses internal standard peptides to set parameters and drive spectral acquisition. This real-time scheme dynamically monitors internal standard peptides, performs spectral matching, and then triggers quantitative analysis of heavy and endogenous peptides [136]. The versatility of PRM is a powerful approach for combining exploration and analysis in quantitative phosphoproteomics.

### 4.3. DIA/SWATH

Sequential window acquisition of all theoretical mass spectra (SWATH-MS) operating in a data-independent acquisition (DIA) mode is an approach designed to combine the advantages of both shotgun and targeted proteomics to achieve reproducibility and consistency in a high-throughput manner [137,138]. This method, like PRM, typically utilizes a Quadrupole-Orbitrap/TOF hybrid as a mass analyzer. In principle, the DIA step acquires the full *m*/*z* range of a selected precursor, followed by fragmentation in sequential small *m*/*z* windows. These acquisitions are continuously repeated to collect information on all detectable precursor and fragment ions within the specified windows [139]. Assay development with DIA/SWATH eliminates the requirement for identifying predetermined sets of ions as in SRM/PRM. Instead, defining mass ranges, precursor isolation window width, and number of MS2 scans per cycle are sufficient for data acquisition [137]. A recent study developed a hybrid DIA workflow with spectral libraries from DDA to quantify 36,350 phosphosites in lung cancer cell lines within 2 h [140]. Although this approach can quantify thousands of phosphopeptides, software development for data analysis is still undergoing improvement to extract phosphopeptide-specific information from available empirical knowledge. A few examples of currently available tools include OpenSWATH and DIA-NN [137,141,142].

## 5. Recent Targeted MS Applications to Elucidate Phosphorylation-Driven Mechanisms

Altered phosphorylation drives processes underlying numerous diseases. Disease-associated phosphorylation cascades can be restricted to specific pathways or, in many cases, involves crosstalk with other pathways. Hence, altered phosphorylation-mediated signaling can include tens to hundreds of proteins. These can potentially be followed and analyzed by SRM-based phosphoproteomics (Table 1). One prime example is the elucidation of phosphorylation-driven mechanisms in the EGFR-MAPK-PI3K-mTOR pathways that can be mapped to individual functional phosphosites (Figure 1B,C) [53,54]. For example, in the MAPK signal transduction cascade, MAPK1/ERK2 and MAPK3/ERK1 are two closely related and often indistinguishable protein isoforms, with corresponding phosphorylation sites at T185/Y187 and T202/Y204, respectively. The rapid activation of MAPK1/3 by dual phosphorylation shows similar trends in phosphorylation across these isoforms and indicates a level of redundancy between MAPK1 and MAPK 3. Ribosomal protein S6 (RPS6) is downstream of the MAPK pathway, integrating proliferating signals from both the MAPK and PI3K-mTOR pathways via S235 and S236 phosphorylation from ribosomal S6 and p70S6 kinases, respectively [54,143]. Consequently, phosphorylated RPS6 is a potential biomarker for the growth state of cancer cells and their drug resistance [144,145]. The ability to resolve RPS6 phosphorylation by two different kinases demonstrates the power of SRM assays.

## 6. Quantitative Phosphoproteomics in Modeling of Signaling Pathways

MS-based instrumentations have revealed extensive phosphorylation modifications in the proteome [29,30,69,95,146,147,148,149]. Though with limited data so far, the quantitative capabilities of SRM/MRM-based phosphoproteomics have demonstrated the ability to reliably measure phosphorylation abundances in signaling pathways (Table 1). These changes in phosphorylation in response to defined perturbations can provide useful data to different research areas from cellular biology to modelling. With the need to further understand how PTMs such as phosphorylation shape cellular function, mathematical models are a promising way to associate specific mechanisms with cellular perturbations [150].

The complexity of phosphorylation-mediated signaling and the subsequent heterogeneity make systems-level models most useful. To highlight some relevant sources of cellular complexities: (1) the presence of potential phosphorylation sites on receptors does not mean all sites can be phosphorylated at the same time, or ever at all [151]. Sometimes, spatially restricted subsets of receptors are not in the same compartment as effector proteins [152,153]; (2) phosphorylated sites can bind to adaptor proteins that protect them from dephosphorylation [154]; (3) there are feedback loops that can modulate pathway phosphorylation activity [155]; and (4) many protein–protein interactions are regulated by phosphorylation states, and this binding interaction is dependent on the concentration of binding partners and interaction affinity [156]. Multiple kinases are often coregulated, rather than acting independently, with patterns that modulate molecular switches which contribute toward the cellular response of a perturbation [29,150,157]. Considering the multifaceted nature of phosphorylation in building mathematical models, we need analytical methods that provide information on the state of individual molecules rather than the average state of receptors. Recent advancements in MS-based targeted phosphoproteomics provide the platform to obtain such relevant experimental information. We can also quantify phosphorylation on receptors and its impact on signaling molecules with high throughput. In addition, quantitative approaches can compartmentalize these changes in organellar fractions such as the mitochondria to establish localization indices for phosphorylation modifications [150].

Predictive modeling in cellular signaling aims to develop mathematical tools that are capable of computing expected outputs from given inputs under defined conditions [158]. With a model, one can either explore the possible outcomes with different structures and parameter regimes or optimize the model structures and parameters towards defined functional phenotypes [25]. However, to understand the dynamics of real biological systems at normal or disease states, one needs to determine and/or estimate the model structure and parameter space under different conditions. The parameter spaces under which real signaling networks operate are only a small subset of the essentially infinite parameter space in which a model could theoretically operate. To constrain model parameters to a relevant set, we can use both biophysical properties (e.g., diffusion and spatial scales) and experimental measurements. For example, for signaling networks the range of stimulation should be constrained to physiologically measured limits and relevant timescales [159]. Phosphoproteomics data can also be useful in determining which processes are operational in a cell [160] and how the activation of different proteins in a pathway are corelated either temporally or through a stimulation dose-range. These correlations can provide a useful constraint for the parameter space of a model [161].

A few studies have undertaken modeling approaches using different approaches of generating phosphorylation data (Table 2). Our group was one of the pioneers of using targeted phosphoproteomics datasets for mathematical modeling applications [104]. In a recent paper, quantitative targeted phosphoproteomics was used to determine the absolute abundance and phosphorylation states of proteins in the MAPK pathway, uncovering the negative feedback regulatory mechanisms of the cascade [53]. In turn, these data were suitable for parametrizing a model to predict the resistance of receptor driven ERK signaling to RAF and MEK inhibition [32].

### Modeling Approaches to Capture Phosphorylation Signaling Complexities

In general, mathematical models are generally useful for: (1) aiding experimental designs to test both simple and complex hypotheses; (2) reconciling discrepancies in experimental data; and (3) consolidating knowledge about a system. Simple models like the partial least squares regression (PLSR) and multiple linear regression (MLR) are based on linear regressions [166]. The data are separated into sets of input and output, and a linear solution that correlates the input to the output is determined. However, these models are unable to capture nonlinear phenomena and coupled effects [167], which can be more pronounced in the disease state.

In complex biological systems, reducing the biochemistry (reactions) of the system to construct a network of ordinary differential equations (ODE) or partial differential equations (PDE) is a common approach [168]. Beginning with an equation for each species reaction written in terms of rate of change, empirical data then provide estimates of values of unknown parameters such as kinetic constants [169]. To simplify these models, certain assumptions are frequently made, for instance multiple phosphorylation sites on a peptide/protein having a similar impact [170]. Including all the phosphorylation states of a protein in a model can lead to a combinatorial explosion [171,172] and an exponential increase in reactions and subsequent model size. For example, a study that modeled signaling in the ERK and Akt network that included 7 receptor dimers and 28 proteins in response to two ligands generated a total of 499 differential equations, 201 reaction rates, and 28 non-zero initial conditions [2. Biological dynamics complicate parameter optimization of ODE/PDE models. One way around the complexity of ODE/PDE models is to simplify the system by “coarse-graining” which reduces the number of equations to just those describing molecular patterns that operate interdependently and are relevant to the system [158,173,174].

Rule-based models offer an approach which factors in the combinatorial complexities of cellular signaling mechanisms [174]. These models are based on the modularity of interactions, representing biomolecular interactions in terms of rules instead of equations [158,175]. These rules then are used to generate reactions and species. The rules are comprised of recognition patterns, reactant-product maps, and rate laws [174]. Consequently, the rules can specify the features of proteins (condition), such as the modification state of a phosphorylation site, which are required for or affected by a particular interaction (action) which can then be executed as simulations. As such, multiple species may qualify as reactants as defined by a rule, and multiple reactions may be generated as defined by the same characteristic law rate [176]. The common languages to write and simulate rule-based models are BioNetGen Language (BNGL) [34] and Kappa [35]. These algorithms run their simulations as: (1) indirect methods—where the state of the system is defined by traditional population variables such as concentrations; (2) direct methods—where the state of the system is defined by the collective states of individual sites or particles; and (3) hybrid particle/population method—where a subset of species can be treated as population variables rather than particles [34,177,178,179]. An example of an application of rule-based models was reported by Chylek et al. A rule-based modeling approach was developed which used temporal global phosphoproteomics data to characterize reshaping of the T-cell phosphoproteome in response to TCR/CD28 co-stimulation. The implicated signaling mechanisms were dependent on specific site phosphorylations and phosphoforms of receptors. The model detected over 100 pTyr sites with greater than two-fold changes in abundance and identified novel negative regulatory sites that were subsequently validated experimentally [165]. Rule-based modeling has also been combined within a Python-based programming language, PySB, which promotes model extensibility and reuse [180].

Rule-based and ODE/PDE models rely heavily, and in some cases entirely, on generated experimental knowledge of well-studied systems [158], which is an opportunity to leverage MS-based phosphoproteomics. MS-based technologies can generate datasets containing site and abundance information that can provide a basis, or “rules” to build biologically realistic models. The ongoing generation of high-quality proteomics and phosphoproteomics data across different experimental sources (such as different cell lines and tissue types) will be essential for the generation of cell-type specific models, and to understand the basis by which such specificity arises [181].

Previous studies have shown the utility of experimental data for constraining models and estimating model parameters. In considering the power of different experimental methods we argue that absolute quantification methods, such as targeted phosphoproteomics, provide much greater advantage in parameter estimation than relative quantification of phosphorylation levels. Absolute quantification can greatly reduce parameter space by directly providing values for protein abundance (concentration) and phosphorylation levels. Most kinetic models use “lumped” rate constants that combine both protein abundance and activity. Since protein abundance varies across cell types [181], knowing this parameter provides a way to infer real differences in protein activity. In addition, functional protein abundance is frequently dependent on protein phosphorylation state. For example, phosphorylation of SOS1 allows binding of 14-3-3 proteins that removes SOS1 from the pool that can participate in signaling [182]. Knowing the functional abundance of proteins through a combinate of targeted proteomics and phosphoproteomics reduces the uncertainty in model parameters and significantly improves model identifiability (Figure 2).

## 7. Future Perspectives

Targeted MS-based quantitative proteomics is becoming increasing popular across scientific and clinical research areas due to significant advancements in assay sensitivity, specificity, and reproducibility. These exceptional characteristics make them suitable for multiple applications such as biomarker discovery in basic and translational research. Classic SRM/MRM-based proteomics has been applied to verify protein biomarkers for different types of diseases such as melanoma, prostate cancer, lung cancer, Parkinson’s disease, cardiovascular diseases, and type 1 diabetes using different specimens like plasma, serum, urine, cerebrospinal fluid, and tissue [183]. Clinical applications could benefit immensely from the implementation of targeted phosphoproteomics methods to identify comprehensive site-specific phosphorylation that are of diagnostic and prognostic assessment value, such as mechanisms of cancer resistance mediated by the PI3K/AKT/mTOR signaling network. The constant improvement and application of SRM-based phosphoproteomics will advance the detection and quantification of biologically relevant phosphorylation sites, which could serve as biomarkers.

However, the broad utility of phosphoproteomics is not without existing challenges. Despite great improvements in proteomics workflows, sample throughput in MS-based proteomics is still relatively low compared to antibody-based techniques, such as Reverse Phase Protein Arrays [184]. Phosphorylation sites that can serve as biomarkers are typically of very low abundance and can be obscured by irrelevant peptides. Fortunately, fast, high-resolution, and highly sensitive measuring instruments like the orbitrap are being integrated into acquisition workflows as in the case of PRM applications. An added benefit with the orbitrap is the potential for simultaneous discovery and targeted phosphoproteomics as demonstrated by DIA/SWATH. Such approaches can significantly reduce instrumentation time, increase throughput, and enable measurements across many samples. DIA-based phosphoproteomics platforms could also expedite deep mining of phosphorylation events in a targeted fashion. Furthermore, the gravitation towards artificial intelligence capabilities like machine learning proposes the utility of phosphoproteomics data to achieve more precise and scalable models in big data analytics. The applications of artificial intelligence to phosphoproteomics data fall in two folds. On one hand, the unsupervised machine learning algorithms where the program is allowed to independently find patterns and trends. On the other hand, the supervised machine learning algorithms can be used to better identify the spectrum from raw MS data as well as interpolate and extrapolate the unmeasured time points and doses with a high fidelity [185]. In either choice, it is important to consider the quality of data that the algorithm learns, because noisy data will not generate meaningful models. Worth noting is that the recently developed physics-informed machine learning methods [186], such as neural differential equations [187], sparse regression [188], and symbolic regression [189], can be further developed and applied in proteomic-data-driven systems identifications and predictions. These applications require more accurate information from measurements, where biophysical rules to be discovered can only operate with absolute quantification of variables in the machine learning models. Besides phosphoproteomics, the algorithms can be designed to integrate other data, such as genomics, epigenomics, transcriptomics, metabolomics, etc. [190].

Another challenge is that due to the low abundance levels of phosphorylation sites, a large amount of input material is usually required for enrichment and subsequent analysis. This could pose a limitation for applications that generate only a small amount of sample/specimen. With the introduction of single-cell proteomics, there is the possibility for pushing the sensitivity of targeted phosphoproteomics to measure target analytes in small amounts of samples, such as those generated by flow cytometry. In terms of specificity, mass spectrometers cannot always differentiate all target phosphopeptides, especially if multiple phosphorylations result in similar mass or structural changes to the peptides. Consequently, utilizing recent tools like structure for lossless ion manipulation (SLIM) in resolution-type analysis, which can distinguish targets by size and shape in addition to mass, may help to resolve this issue [191]. Furthermore, bioinformatic tools are being developed that can identify functionally important phosphorylation sites within a context of a multi-phosphorylated protein. These are based on deep-learning programs like AlphaFold2 [192] that can potentially predict protein structural changes arising from phosphorylation events. Combining advanced MS and bioinformatic tools could benefit SRM applications for site specificity and help overcome ambiguous site-specific identifications using traditional MS approaches.

### Concluding Remarks

As biomedical research gravitates towards a systems approach to answer important and increasingly complex questions, the integration of experimental and computational tools will continue to increase. To support this work, targeted phosphoproteomics has provided a platform to obtain high quality data with scalable throughput. It also provides important information for building predictive models that can assist in experimental designs and clinical decision making. Experimental and computational integration hold promise to better decipher phosphorylation mechanisms underlying disease pathogenesis and identify potential biomarker/therapeutic candidates for use in traditional and precision medicine.

## Figures and Tables

**Figure 1 molecules-28-01143-f001:**
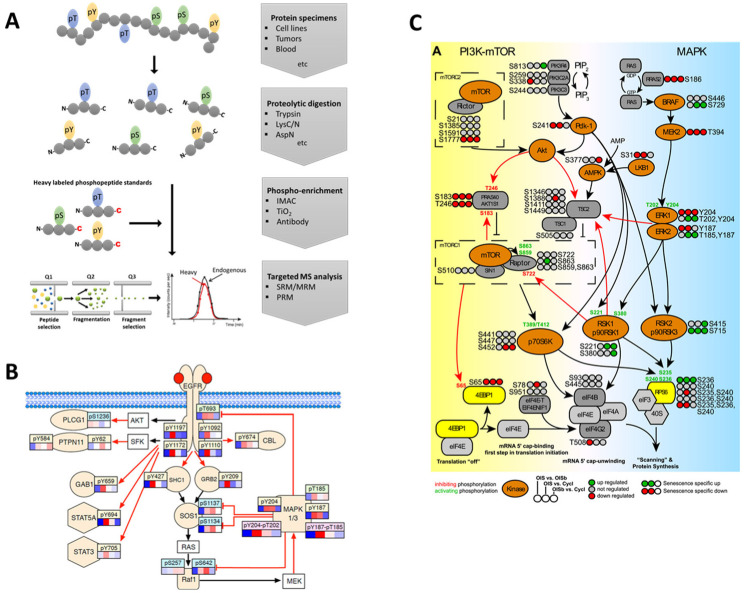
(**A**) Standard workflow for targeted analysis of protein phosphorylation. Selected studies have demonstrated the use of SRM to measure phosphorylation dynamics of signaling molecules complexes such as the (**B**) EGFR-MAPK pathway [53] and (**C**) PI3K-mTOR and MAPK pathway [54].

**Figure 2 molecules-28-01143-f002:**
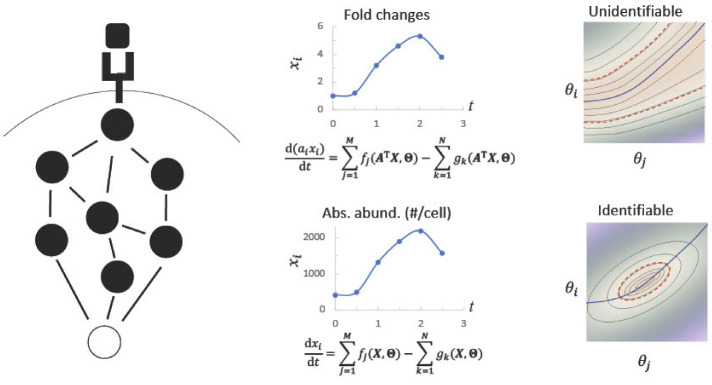
Absolute quantifications increase the identifiability of model parameters.

**Table 1 molecules-28-01143-t001:** Select studies that have demonstrated the use of targeted phosphoproteomics to quantify phosphorylation dynamics of signaling pathways.

Pathway	Goal	Sample Type	Starting Amount	Enrichment Strategy	Quantified Phosphosites	Reference
Breast cancer	Classify high- and low-risk patient groups	Breast tumor	500 µg	Fe^3+^-IMAC	19	[124]
PI3K-mTOR and MAPK	Oncogene-induced senescence in the presence of BEZ235, a dual PI3K/mTOR inhibitor	Human diploid fibrolast (HDF) cell line	200 µg	Ti^4+^-IMAC	89	[54]
PI3K-mTOR and MAPK	Phosphorylation dynamics of rapamycin mechanistic targets	PC9 and H1975 NSCLC cell line	300 µg	Ti^4+^-IMAC + HpH	42	[125]
PI3K-mTOR and Cell Cycle	Sensitive disruptions MAPK, PIK3/mTOR, and cell cycle signaling pathways using 26 inhibitors	MCF7, PC3, and HL60 cell lines	500 µg	Fe^3+^-IMAC	92	[126]
DNA damage response (DDR)	Understanding DDR functionality, in response to ionizing radiation and methyl methanesulfonate	MCF10A cell line and human PBMCs	200 µg	Fe^3+^-IMAC	107	[127]
DNA damage response (DDR)	Understanding DDR functionality, in response ionizing radiation	MCF10A cell line and human breast tumors	500 µg	Antibody	29	[128]
DNA damage response (DDR)	Understanding DDR functionality, in response ionizing radiation and 4-nitroquinoline 1-oxide	LCL cell line and human PBMCs	500 µg	Antibody	25	[129]
EGFR-MAPK	Phosphorylation dynamics in response to EGF perturbation	MCF7 and Hs578T cell lines	25–100 µg	Fe^3+^-IMAC and TiO_2_	34	[53]
MAPK	Rewiring of MAPK signaling in drug-resistant BRAF^V600E^ melanomas	A375 cell line	10 and 100 µg	Fe^3+^-IMAC	22	[32]
Multiple cancer pathways	Characterize dynamic signaling across diverse cancer pathways	Patient-derived xenografts of triple negative breast cancer and human medulloblastoma tumors	500 µg	Antibody and Fe^3+^-IMAC	284	[93]

**Table 2 molecules-28-01143-t002:** Selected applications of phosphorylation data in mathematical modeling.

Pathway	Data Source	Experimental Conditions	Model	Readouts	Literature
ErbB signaling pathway	Western blotting, literature	10 timepoints, 2 ligands, 2 concentrations, 4 cell lines	ODE	3 phosphorylated proteins	[162]
Multisite Tau phosphorylation	Western blotting	5 timepoints, 2 ligands	ODE	10 phosphosites from 2 phosphorylated proteins	[163]
MAPK signaling	Western blotting	2 timepoints, 2 ligands, 3 gene knockdowns	Modular Response Analysis	3 phosphorylated proteins	[164]
T Cell receptor signaling	Global Phosphoproteomics	5 timepoints, 2 stimulators	NFSim (rule-based model)	22 phosphosites from 17 phosphorylated proteins	[165]
MAPK signaling	ELISA and targeted phosphoproteomics	1 ligand, 2 inhibitors at 10 concentrations	MARM1 (rule-based model)	2 phosphorylated proteins	[32]

## Data Availability

No new data were created or analyzed in this review. Data sharing is not applicable.

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
