# Peer review of "Targeted Quantification of Protein Phosphorylation and Its Contributions towards Mathematical Modeling of Signaling Pathways"

_molecules, 2023, doi:10.3390/molecules28031143_

Round 1

Reviewer 1 Report

This is a relatively comprehensive, informative, well-considered, well-structured and well-organized review paper that provides a comprehensive and systematic introduction to the general processes and data models of protein phosphorylation, which helps researchers studying protein phosphorylation to gain a better understanding in this field.  It can be inferred that the authors are familiar with the field of protein phosphorylation and have a clear understanding of the advantages and shortcomings of this technology, as well as the inclusion and description of some emerging technologies in recent years, and overall this review is recommended for publication. Some of my questions are as follows:

(1)    How to perform deeper mining and database supplementation of protein phosphorylation data?

(2)    Cross-referencing of protein phosphorylation data from different research areas, such as analysis and prediction of common pathways in protein phosphorylation studies in medicine, agronomy, food, biology and other disciplines? What is your better way?

(3)    What do you know about how to apply artificial intelligence and big data to analyze models?

Author Response

The authors would like to thank the reviewer for his comments and questions, which have helped to contribute to the resourcefulness of the manuscript. Attached are the responses to the questions raised.

Reviewer 2 Report

The review article  “Targeted quantification of protein phosphorylation and its contributions towards mathematical modeling of signaling pathways”, Dakup et al discussed about Mass Spectrometry (MS)-based targeted proteomics to quantify relative and absolute abundances of protein phosphorylation. While this topic is of interest, the article has superficial content and needs detailed explanation. The article is unacceptable in its present form because author should address the following concerns:

Comment1: Author mention used of different proteolytic enzymes for digestion. For digestion selection of substrate :enzyme ratio is crucial step. Author include the how varying conc. of substrate: enzyme impact the phospho-protein identification.

Comment2: Author should include impact of sequential digestion on coverage and identification of phosphosites.

Comment3: Author should mention the criteria for selection of precursor ions (intensity threshold, number of amino acid per peptide) and fragmentation ions (number of b and y ions per peptides) for targeted acquisition.

Comment4: The most of time the standard peptides (AQUA) interferes with protein spectra, author should provide how to choose optimal concentration of internal standards for spiking.

Comment5: Author should include the normalization strategy for identified peptides with respect to total protein in complex mixture and for pure proteins.

Comment6: In Fig 1(A), Author should mention software used for targeted data analysis (Skyline, Scaffold DIA etc).

Author Response

The authors would like to thank the reviewer for his insights into technical aspects of the manuscript to improve the overall quality. Attached are the responses to the questions raised.

Round 2

Reviewer 2 Report

The author has addressed the all raised concerned and made the changes accordingly.